# Role of TLR4 Receptor Complex in the Regulation of the Innate Immune Response by Fibronectin

**DOI:** 10.3390/cells9010216

**Published:** 2020-01-15

**Authors:** Mingzhe Zheng, Anthony Ambesi, Paula J. McKeown-Longo

**Affiliations:** Department of Regenerative & Cancer Cell Biology, Albany Medical College, Albany, NY 12208-3479, USA; mingzhe.zheng9@gmail.com (M.Z.); ambesia@amc.edu (A.A.)

**Keywords:** fibronectin, TLR4, fibrosis, inflammation, IL-8, CD14

## Abstract

Chronic inflammation and subsequent tissue fibrosis are associated with a biochemical and mechanical remodeling of the fibronectin matrix. Due to its conformational lability, fibronectin is considerably stretched by the contractile forces of the fibrotic microenvironment, resulting in the unfolding of its Type III domains. In earlier studies, we have shown that a peptide mimetic of a partially unfolded fibronectin Type III domain, FnIII-1c, functions as a Damage Associated Molecular Pattern (DAMP) molecule to induce activation of a toll-like receptor 4 (TLR4)/NF-κB pathway and the subsequent release of fibro-inflammatory cytokines from human dermal fibroblasts. In the current study, we evaluated the requirement of the canonical TLR4/MD2/CD14 receptor complex in the regulation of FnIII-1c induced cytokine release. Using dermal fibroblasts and human embryonic kidney (HEK) cells, we found that all the components of the TLR4/MD2/CD14 complex were required for the release of the fibro-inflammatory cytokine, interleukin 8 (IL-8) in response to both FnIII-1c and the canonical TLR4 ligand, lipopolysaccharide (LPS). However, FnIII-1c mediated IL-8 release was strictly dependent on membrane-associated CD14, while LPS could use soluble CD14. These findings demonstrate that LPS and FnIII-1c share a similar but not identical mechanism of TLR4 activation in human dermal fibroblasts.

## 1. Introduction

Chronic inflammation plays a significant role in many fibrotic diseases including cancer. Most solid tumors are characterized by an infiltration of fibroblasts, which under the influence of the tumor cells, differentiate into highly contractile myofibroblasts. The generation of the myofibroblast phenotype is accompanied by increases in both the fibronectin matrix and in the mechanical forces placed upon it. The signaling networks between stromal and cancer cells are exceedingly complex and interdependent, occurring on a background of poorly understood mechanical signals which are activated in response to increasing tissue rigidity. The tumor microenvironment is characterized by fibrosis and inflammation which contributes to tissue rigidity and is considered integral to tumor growth and metastasis [1]. 

Fibronectin is an extracellular matrix (ECM) protein which is polymerized by adherent cells into a mechanically sensitive network of interacting fibers. Fibronectin is up-regulated in the stroma of solid tumors and has been shown to contribute to cancer cell growth, migration, invasion, survival and resistance to chemotherapy [2,3]. Consequently, the molecular pathways activated by stromal fibronectin are now regarded as potential drug targets [4]. However, the molecular pathways regulated by the pathological remodeling of stromal fibronectin are not well understood. Structurally, the fibronectin molecule consists of independently folded domains termed Type I, II, and III based on shared amino-acid homologies. Polymerized fibronectin fibers are conformationally labile and respond to force by unfolding their Type III domains, which unlike the Type I and II domains, are not stabilized by disulfide bonds [5]. The unfolding of the Type III domains can cause fibronectin to stretch up to 8 times its length [6,7]. Studies have now demonstrated fibronectin in the stroma of solid tumors to be highly stretched due to the unfolding of Type III domains [8,9,10]. The impact of this strained form of fibronectin on cancer progression is not known. 

To understand the potential impact of fibronectin strain on tumor progression, we have used a fibronectin peptide, FnIII-1c, which corresponds to a stable intermediate structure predicted to form during force induced unfolding of the first Type III domain of fibronectin [11]. We have identified this peptide as a Damage Associated Molecular Pattern molecule or DAMP which induces the expression of several fibro-inflammatory genes in human dermal fibroblasts [12,13,14], DAMPs are endogenous products of tissue damage which work through toll-like receptors (TLR) to activate innate immune responses [15]. DAMPs arise early during tumor progression as the ECM is actively remodeled [16]. TLRs are a family of transmembrane receptors which were first identified on immune cells, as initiators of the innate immune response to pathogens, such as the bacterial cell wall component, LPS [17]. TLRs have also been identified on other cell types including fibroblasts, epithelial cells, endothelial cells and tumor cells [18,19,20,21]. TLRs function in complexes with co-receptors and ancillary proteins whose specific functions are best understood for the activation of TLR4 by its prototype ligand, the Pathogen Associate Molecular Pattern molecule or PAMP, LPS [22]. TLR4 activation in response to LPS requires two accessory molecules, CD14 and MD2 [23]. MD2 is a secreted protein which complexes with TLR4, binds the Lipid A moiety of LPS, and facilitates the formation of TLR4 dimers which are required for downstream signaling and activation of NF-κB [23]. CD14 is GPI-linked protein found on the cell membrane in lipid rafts and functions to transfer LPS from the bacterial cell wall to the MD2/TLR4 complex [24]. CD14 can also be lost from the cell surface by both protease dependent and independent shedding [25]. Shed CD14 is found in a soluble form in most body fluids including blood plasma and serum [26]. Soluble CD14 can also complex with LPS to initiate a TLR4 dependent inflammatory response [27]. FnIII-1c elicits expression of several fibro-inflammatory genes in dermal fibroblasts including IL-8 and tumor necrosis factor-alpha (TNF-α). Cytokine release in response to FnIII-1c depends on a TLR4/NF-κB signaling pathway [12,13]. A variety of matrix-derived DAMPs have been identified in solid tumors including proteins, proteoglycans, and glycosaminoglycans. These DAMPs work primarily through a TLR4/NF-κB pathway to promote chronic inflammation and to drive the expression of growth and survival genes all of which are thought to contribute to tumor progression (reviewed in [28,29]). The molecular basis underlying the recognition of such a diverse array of ligands is not well understood. In some instances, TLRs can function in complexes with specific accessory proteins or co-receptors, thereby achieving ligand specific outcomes [30,31].

In the current study we evaluate the contribution of the components of the TLR4 receptor complex in the induction of cytokines by LPS and FnIII-1c. The data indicate that the fibronectin DAMP, FnIII-1c, similar to LPS, requires the TLR4/MD1/CD14 receptor complex to induce cytokine release from dermal fibroblasts. In contrast to LPS, FnIII-1c has a strict requirement for the membrane-bound form of CD14. These data suggest that targeting membrane-associated CD14 may provide a novel approach to designing therapeutics directed at controlling DAMP-dependent fibro-inflammatory disease.

## 2. Materials and Methods

### 2.1. Materials

Ultrapure LPS purified from E. coli K12 strain was purchased from InvivoGen (San Diego, CA, USA). Human IL-8 enzyme-linked immunosorbent assay (ELISA) kit was from Becton Dickinson Biosciences (San Jose, CA, USA). Recombinant human CD14, human TNF-α, human IL-1α, anti-human MD-2 antibody, and neutralizing antibodies: anti-human CD14, anti-human TLR2 and anti-human TLR4 were purchased from R&D Systems (Minneapolis, MN, USA). DL-Sulforaphane and phosphatidylinositol specific phospholipase C (PI-PLC) were purchased from Sigma-Aldrich-Millipore (St. Louis, MO, USA) TAK-242 was from Calbiochem/EMD Millipore (Billerica, MA, USA). 

### 2.2. Recombinant Fibronectin Modules

The recombinant fibronectin Type III modules were generated through polymerase chain reaction amplification of the human fibronectin cDNA clone pFH100 as previously described [32]. DNA encoding fibronectin amino acids Asn-631 to Pro-705 for III1c or Asn-1904 to Thr-1992 for III13 was cloned into vector pQE-70 (Qiagen, Germantown, MD, USA) and expressed in M15 bacteria. Recombinant proteins were purified by sequential chromatography with metal-chelating nitrilotriacetic acid agarose (Ni-NTA), sephadex G-25, and ion exchange columns as previously described [33].

### 2.3. Stably Transfected HEK 293 Cell Lines

The pDUO-hTLR4A/MD2 (a pDUO plasmid co-expressing the human TLR4A and MD2 genes from InvivoGen) or pDUO-hTLR4A/CD14 plasmid (a pDUO plasmid co-expressing the human TLR4A and CD14 genes from InvivoGen (San Diego, CA, USA), were transfected into HEK 293 cells using Lipofectamine 2000 (Invitrogen/Life Technologies Corp., Grand Island, NY, USA). Two days later, cells were placed in selection medium (culture medium containing 10 µg/mL blasticidin). Clones were isolated from blasticidin-resistant cells by serial dilutions. Protein expression was screened by immunoblot analysis. A clone of 293-hTLR4A-MD2 and a clone of 293-hTLR4A-CD14 were selected for experiments. The other stably transfected HEK 293 cell lines: 293-Null, 293-hTLR4A, 293-hTLR4A-MD2-CD14, and 293-hMD2-CD14 were purchased from InvivoGen (San Diego, CA, USA).

### 2.4. Cell Culture

Human dermal fibroblasts were grown in Dulbecco’s modified eagle medium (DMEM) (Invitrogen/Life Technologies, Corp., Grand Island, NY, USA) containing 50 U/mL penicillin, 50 mg/mL streptomycin, and 10% FBS (HyClone Laboratories/GE Healthcare Life Sciences, Logan, Utah, USA). 293-hTLR4-MD2-CD14 cells were maintained in DMEM containing 10 µg/mL blasticidin, 50 µg/mL of HygroGold, 50 U/mL penicillin, 50 mg/mL streptomycin and 100 µg/mL normocin. In addition, all other transfected HEK293 cells were maintained in DMEM containing 10 µg/mL blasticidin, 50 U/mL penicillin, 50 mg/mL streptomycin and 100 µg/mL normocin. 

### 2.5. Cell Treatments

Protease-free bovine serum albumin (BSA) from Roche Diagnostics (Indianapolis, IN, USA) was further processed for cell culture usage. Under sterile condition, 10% BSA in phosphate buffered saline (PBS) was dialyzed vs PBS containing 100 µg/mL Penicillin-Streptomycin (pen-strep) overnight with a dialysis tube (12–14 kDa molecular weight cut off) for 1 day and further vs DMEM containing pen-strep overnight. Dialyzed BSA solution’s volume was measured and the solution was sterilized with 0.22 µm syringe filter. BSA solution was inactivated by incubation at 56 °C for 1 h. 

For experiments, cells were placed in DMEM containing 10% FBS, 50 U/mL penicillin, 50 mg/mL streptomycin and 10 mM HEPES (10% FBS/DMEM), and DMEM containing 0.1% BSA, 50 U/mL penicillin, 50 mg/mL streptomycin, 1× non-essential amino acids and 10 mM HEPES (0.1% BSA/DMEM). Human dermal fibroblasts were plated onto 48-well culture plates (2 × 10^4^ cells per well) and cultured overnight. When inhibitors, blocking antibodies or PI-PLC were used, cells were pre-incubated for the indicated times. For serum-free conditions, human dermal fibroblast monolayers were washed twice and incubated with 0.1% BSA/DMEM overnight prior to treatments. Transfected HEK293 cells were plated onto collagen-coated 48-well culture plates (2 × 10^4^ cells per well) and cultured for two days in the presence of selection antibiotics. Cells were treated with fibronectin modules or LPS without selection antibiotics. In serum-free condition, transfected HEK293 monolayers were gently washed once and incubated with 0.1% BSA/DMEM for two hours prior to treatments.

### 2.6. Human IL-8 Enzyme-Linked Immunosorbent Assay

After cell treatment, condition medium was collected and centrifuged. Supernatant was saved and properly diluted with PBS containing 10% FBS. IL-8 protein concentration in condition medium was determined by using a human IL-8 ELISA kit (BD Biosciences, San Diego, CA, USA). 

## 3. Results

### 3.1. Cytokine Induction in Response to Either FnIII-1c or LPS is Dependent on TLR4

To evaluate the TLR receptor complex on dermal fibroblasts which regulates cytokine release in response to either pathogens or endogenous DAMPs, human dermal fibroblasts were incubated with increasing amounts of either the unfolded Type III-1 domain of fibronectin (FnIII-1c) or LPS. As we have previously reported [12,13], incubation of human dermal fibroblasts with FnIII-1c resulted in an increase in the synthesis of IL-8 (Figure 1A). The effect of FnIII-1c on cytokine expression was dose-dependent reaching over 30 ng/mL within 24 h. A second fibronectin Type III domain, FnIII-13, did not elicit an IL-8 response and served as negative control. LPS also induced a dose response increase in IL-8 reaching over 100 ng/mL within 24 h (Figure 1B). Induction of IL-8 by either FnIII-1c or LPS was completely inhibited using a blocking antibody to TLR4. Neither control IgG nor blocking antibody to TLR2 had any effect on IL-8 production in response to either FnIII-1c or LPS (Figure 1C). The inhibitor of downstream signaling from TLR4, TAK-242, also prevented the release of IL-8 in response to both FnIII-1c and LPS. In contrast, TAK-242 had only a slight effect on the synthesis of IL-8 in response to TNFα which does not depend on TLR4 signaling (Figure 1D). These data indicate that TLR4 signaling mediates cytokine release from dermal fibroblasts in response to both the bacterial pathogen LPS and the matrix-derived DAMP, FnIII-1c. 

### 3.2. TLR4 Signaling in Response to Either FnIII-1c or LPS Depends on MD2 and CD14

Activation of TLR4 signaling on immune cells by LPS is known to depend on MD2. To determine whether MD2 is required for the activation of TLR4 signaling on dermal fibroblasts, cells were pre-incubated with sulforaphane, an inhibitor of MD2. As shown in Figure 2A, sulforaphane inhibited the synthesis of IL-8 in response to both LPS and FnIII-1c. In contrast, sulforaphane did not block IL-8 induction in response to TNF-α. 

CD14 is a GPI-linked cell surface protein, typically found in lipid rafts in the plasma membrane where it binds LPS and facilitates its transfer to the TLR4/MD2 complex to initiate TLR signaling. To evaluate the role of CD14 in LPS and FnIII-1c mediated cytokine release by dermal fibroblasts, cells were incubated with either LPS or FnIII-1c and increasing concentrations of blocking antibody to CD14. Under these conditions, the stimulation of IL-8 production by either LPS or FnIII-1c was completely inhibited by anti-CD14 antibody. Taken together, these data indicate that induction of IL-8 in dermal fibroblasts by both LPS and FnIII-1c requires activation of the TLR4/MD2/CD14 signaling complex. 

### 3.3. IL-8 Induction by LPS Is Serum-Dependent in Dermal Fibroblasts

In addition to its location on the plasma membrane, CD14 is also found in a soluble form in serum [34]. Both soluble and membrane CD14 have been reported to facilitate the activation of TLR4 by LPS. To evaluate the contribution of serum-derived CD14 in the release of cytokines by dermal fibroblasts, cells were incubated with LPS in the presence or absence of serum. Figure 3A shows that in the absence of serum, fibroblasts were completely unresponsive to LPS, even at relatively high concentrations (1 µg/mL). In the presence of 10% serum, LPS elicited a dose-dependent increase in IL-8 release. In contrast to LPS, FnIII-1c stimulated a dose-dependent release of comparable levels of IL-8 in the presence or absence of serum (Figure 3B). These results suggest that LPS but not FnIII-1c has a serum requirement to induce synthesis of IL-8 in dermal fibroblasts. Taken together with the observation that TLR4 signaling in response to LPS depends on CD14 (Figure 2B), the data suggest that the LPS-mediated synthesis of IL-8 by dermal fibroblasts depends on the soluble CD14 present in serum. To confirm that soluble CD14 supports the induction of IL-8 by LPS, fibroblasts were incubated with either LPS or FnIII-1c in the presence of exogenous purified soluble CD14. In the absence of exogenous CD14, fibroblasts synthesized very little IL-8 over a 24 hr period in response to LPS. In contrast, treatment with Fn III-1c resulted in the release of over 20 ng/mL of IL-8 (Figure 3C). When serum-free medium was supplemented with increasing concentrations of exogenous CD14, there was a dose-dependent increase in IL-8 synthesis in response to both FnIII-1c and LPS. The data are consistent with a role for CD14 in the initiation of TLR4 signaling in response to both LPS and FnIII-1c, but indicate that fibroblasts respond poorly to LPS unless supplemented with an exogenous source of CD14. In contrast, FnIII-1c elicited a robust cytokine response in the absence of exogenous CD14, indicating that membrane-bound CD14 is sufficient to initiate TLR4 signaling in dermal fibroblasts in response to FnIII-1c.

### 3.4. IL-8 Induction by FnIII-1c Requires Membrane-Bound CD14 in Dermal Fibroblasts

The data indicate that dermal fibroblasts do not response to LPS unless provided with an exogenous source of CD14. In contrast, dermal fibroblasts do respond to FnIII-1c, suggesting that membrane-bound CD14 is sufficient to activate TLR4 signaling in response to FnIII-1c but not LPS. CD14 is attached to the plasma membrane by a glycophosphatidylinositol (GPI) linkage and can be removed from the membrane by treatment with the enzyme, Phosphatidylinositol-phospholipase C (PI-PLC). To evaluate further the requirement for membrane-bound CD14 in mediating cytokine release in response to FnIII-1c, dermal fibroblasts were pretreated with PI-PLC prior to incubation with either FnIII-1c or LPS in the presence of serum. TNF and IL-1 treated cells served as a positive control for IL-8 induction as the receptors for these molecules do not depend on CD14 for signaling. As shown in Figure 4A, pretreatment of cells with increasing amounts of PI-PLC resulted in a complete inhibition of IL-8 secretion in response to FnIII-1c, while having little effect on the response to LPS. Induction of IL-8 in response to either TNF or IL-1 was largely unaffected by enzyme treatment. The data are consistent with data in Figure 3 showing that the CD14 present in the serum is sufficient to activate TLR4 signaling in response to LPS. In contrast, enzyme treatment prevented activation of TLR4 in response to FnIII-1c consistent with FnIII-1c requiring membrane-bound CD14. 

The requirement for membrane-bound CD14 in TLR4 activation was also assessed using HEK-293 cells engineered to express various components of the TLR4/MD2/CD14 receptor complex. HEK-293 cells, which do not express any components of the TLR4 receptor complex, were transfected with the genes for TLR4, MD2, or CD14 individually and in combination. The cells expressing various combinations of receptor components were then incubated with either LPS or FnIII-1c in the presence of serum. As shown in Figure 4B, HEK-293 cells expressing only TLR4 or TLR4-CD14 did not release IL-8 in response to either LPS or FnIII-1c, confirming a requirement for MD2 in the activation of TLR4 by these ligands. LPS was able to induce IL-8 secretion in HEK-293 cells expressing only TLR4 and MD2. This finding is consistent with an earlier observation that LPS does not require membrane-bound CD14 initiate TLR4 activation (see Figure 3A and Figure 4A). In contrast, FnIII-1c did not elicit an IL-8 response under these conditions. However, in HEK-293 cells where CD14 was also expressed, FnIII-1c did induce IL-8 secretion, consistent with the membrane form of CD14 being required for the activation of TLR4 signaling by FnIII-1c (Figure 4B). 

Treatment of HEK-293 cells expressing TLR4/MD2 or TLR4/MD2/CD14 with increasing amounts of LPS resulted in a robust induction of IL-8 in both cell types when treatments were performed in the presence of 10% serum (Figure 5A). In the absence of serum, cells lacking CD14 were non-responsive to LPS (Figure 5B). In contrast, FnIII-1c was unable to elicit an IL-8 response in HEK-293 cells expressing TLR4/MD2 in either the presence or absence of serum (Figure 5C,D), suggesting that soluble CD14 cannot support TLR4 activation by FnIII-1c. Consistent with this prediction, incubation of FnIII-1c with HEK-293 cells expressing TLR4/MD2 did not induce IL-8 even when supplemented with purified soluble CD14. In contrast LPS was able to elicit an IL-8 response under these conditions (Figure 5E). These experiments indicate that in dermal fibroblasts both MD2 and CD14 are required for the TLR4 dependent induction of cytokines by either LPS or FnIII-1c. The data also show that while both FnIII-1c and LPS require CD14 for TLR4 activation, there is a strict requirement for cell surface membrane-bound CD14 in the activation of TLR4 by FnIII-1c, while LPS can use the soluble form of CD14 to initiate signaling.

## 4. Discussion

Sterile inflammation occurs when DAMPs are released into the extracellular microenvironment in response to injury or during the progression of fibro-inflammatory associated pathologies. DAMPs can promote wound healing by regulating the innate immune response during the inflammatory phase of wound repair [15]. DAMPs are recognized by TLR receptor complexes which activate the NF-κB dependent expression of inflammatory and profibrotic genes. When this process becomes dysregulated, feed forward loops are created which lead to chronic inflammation and eventual tissue fibrosis [35,36,37]. In the case of solid tumors, ECM-derived DAMPs are created when paracrine signaling between the tumor and the stromal fibroblasts results in increased matrix proteolysis, changes in expression of ECM genes and exposure of cryptic sites within the matrix which arise from increased contractile force generated from myofibroblasts [28,38]. DAMPs are comprised of structurally diverse molecules which are derived from ECM as well as from damaged cells [39]. The molecular basis for the recognition of these DAMPs by TLRs remains largely unknown [40]. 

In the current study, we present a side-by-side comparison between the fibronectin-derived DAMP, FnIII-1c, which is known to activate TLR4 signaling [12,13] with the canonical TLR4 ligand, LPS. Our data indicate that both ligands induce the expression of the cytokine and IL-8 in dermal fibroblasts, and that IL-8 expression is prevented by inhibitors of TLR4, MD2, and CD14. These findings indicate that as with LPS, FnIII-1c induces IL-8 through the activation of the TLR4/MD2/CD14 complex. While this is not unexpected, it should be noted that not all ECM-derived DAMPs use CD14 or MD2 to activate TLR4. Hyaluronan is a large molecular weight glycosaminoglycan which is found in the ECM. Low molecular weight hyaluronan which is generated by hyaluronidase in response to injury binds the transmembrane protein, CD44 which then complexes with TLR4 in membrane rafts to activate downstream signaling [41]. Another ECM protein, Tenascin C is up-regulated following injury and during tumor progression. Tenascin C also serves as a TLR4 agonist to promote expression of fibro-inflammatory genes. Activation of TLR4 signaling by Tenascin C does not require either MD2 or CD14 [42]. A recent study in macrophages has shown that the activation of the TLR4/NF-κB pathway by Tenascin C or LPS results in the generation of different macrophage phenotypes characterized by different subsets of activated signaling molecules and the expression of distinct genes [43]. These data suggest that the innate immune response is tailored to the specific DAMP or PAMP present in the tissue microenvironment. The molecular basis of this distinction is likely linked to the use of ligand specific TLR4 co-receptors and adaptor proteins which can discriminate among the various DAMPs present in the tissue. 

The present study documents a distinguishing characteristic of TLR4 activation by the fibronectin DAMP, FnIII-1c. FnIII-1c dependent IL8 secretion in dermal fibroblasts required the membrane-bound species of CD14. Removal of membrane-bound CD14 with phospholipase C, rendered the cells unresponsive to FnIII-1c, even in the presence of exogenous soluble CD14. In contrast, LPS could not induce IL-8 secretion in dermal fibroblasts unless supplemented with an exogenous source of CD14. The requirement for soluble CD14 to initiate TLR4 activation by LPS is not clear but may result from insufficient levels of membrane-bound CD14 on the surface of dermal fibroblasts. Previous studies have suggested that soluble CD14 may be required for TLR4 activation of CD-14 deficient cells such as endothelial or epithelial cells [44]. Nevertheless, FnIII-1c induced cytokine expression in dermal fibroblasts, suggesting there was sufficient CD14 to activate TLR4 in response to FnIII-1c. A similar finding was seen in HEK cells engineered to express TLR4/MD2 where LPS but not FnIII-1c could elicit an IL-8 response in the presence of serum while FnIII-1c induced IL-8 only when cells also expressed membrane-bound CD14. The inability of FnIII-1c to use soluble CD14 suggests that unlike LPS, FnIII-1c may not bind directly to CD14, but may require another as yet unidentified receptor which subsequently interacts with membrane-bound CD14. These data demonstrate that LPS and FnIII-1c activate TLR4 signaling by a mechanism that is only partially shared. The significance of the requirement for membrane-bound CD14 in the initiation of TLR4 signaling by FnIII-1c is not known. Such a requirement places an extra level of control on the immune response of fibroblasts, restricting it to those cells expressing the appropriate co-receptor required for TLR4 activation or to those cells closely associated with strained fibronectin fibers.

The conformational flexibility of fibronectin allows it to respond to changes in tissue stiffness and mechanical force within the tumor microenvironment. Fibronectin has several mechanically regulated activities including polymerization of soluble fibronectin [45,46] as well as the binding of integrins [10], collagen [47], growth factors [48], cytokines [49] and bacterial adhesins [50]. The stromal tissue of squamous cell lung cancer is enriched for fibronectin, myofibroblasts, and IL8, supporting a model of myofibroblast generated contractile force placing fibronectin under sufficient strain to trigger the activation of TLR4 signaling [51]. In a recent study using a 3D model of tumor-associated ECM, we have shown that fibronectin in the tumor-associated ECM activates TLR4 signaling and IL-8 release in lung cancer cells (Cho et al., manuscript under review). 

The proposed activation of TLR4 and subsequent release of IL-8 in response to strained fibronectin present in the tumor stroma is significant. Many solid tumors show high levels of IL-8, a fibro-inflammatory cytokine, which promotes angiogenesis, tumor cell proliferation, metastasis and drug resistance [52,53,54]. Targeting IL-8 has been proposed as a treatment for several cancers including breast cancer [55,56], pancreatic [57], and pre-leukemic stem cells [58]. In patients with chronic obstructive pulmonary disease, IL-8 levels can be diagnostic of lung cancer [59] and serum IL-8 levels are correlative with lung cancer risk [60]. Understanding the molecular steps controlling the mechanically activated signals generated in the tumor ECM will help in the identification of a novel class of therapeutic targets which can be used in the treatment of drug resistant, metastatic tumors. 

## Figures and Tables

**Figure 1 cells-09-00216-f001:**
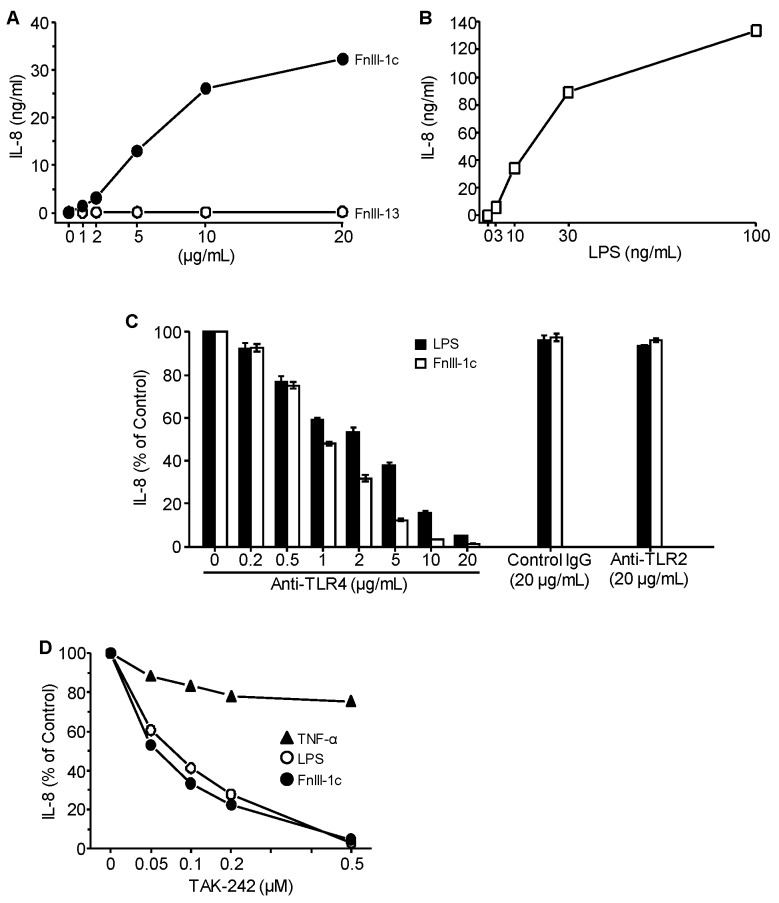
TLR4 mediates IL-8 expression in response to FnIII-1c and LPS in dermal fibroblasts. Monolayers of human dermal fibroblasts in 10% FBS/DMEM were treated for 24 h with (**A**) FnIII-1c or FnIII-13 (1-20 µg/mL), (**B**) LPS (1-100 ng/mL), (**C**) LPS (100 ng/mL) or FnIII-1c (10 µM) in the presence of the designated amounts of blocking antibody to TLR4 or TLR2. IgG served as control. (**D**) TNF-α (25 ng/mL), LPS (100 ng/mL) or FnIII-1c (10 µM) in the presence of increasing amounts of the TLR4 inhibitor, TAK-242. The wells without antibodies (**C**) or inhibitors (**D**) were set as 100%. IL-8 concentration in conditioned medium was determined by ELISA. The data represent the mean ± S.E. of triplicate assays from three separate experiments.

**Figure 2 cells-09-00216-f002:**
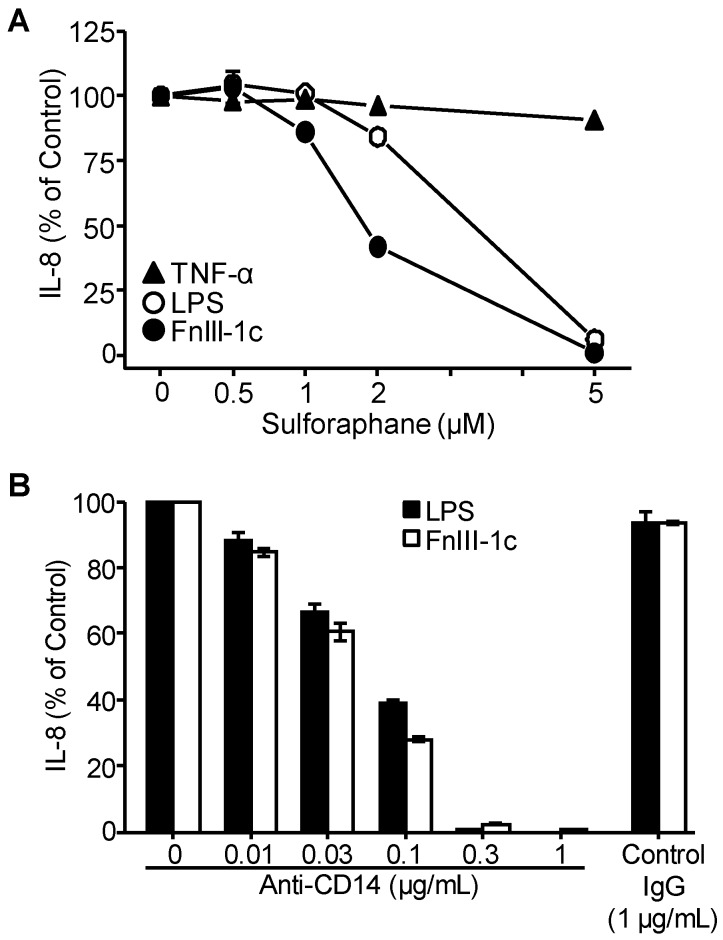
IL-8 induction by LPS and FnIII-1c in dermal fibroblasts requires MD2 and CD14. Monolayers of human dermal fibroblasts were incubated with the (**A**) MD2 inhibitor, sulforaphane, in 10% FBS/DMEM or (**B**) CD14 blocking antibody or control normal IgG in 1% human serum/DMEM. Cells were then treated for 24 hours with either FnIII-1c (10 µM), LPS (100 ng/mL), or TNF-α (25 ng/mL) in the continuing presence of antibodies or inhibitors. IL-8 concentration in conditioned medium was determined by ELISA. The wells without antibodies or inhibitors were set as control (100%). The data represent the mean ± S.E. of triplicate assays from two separate experiments.

**Figure 3 cells-09-00216-f003:**
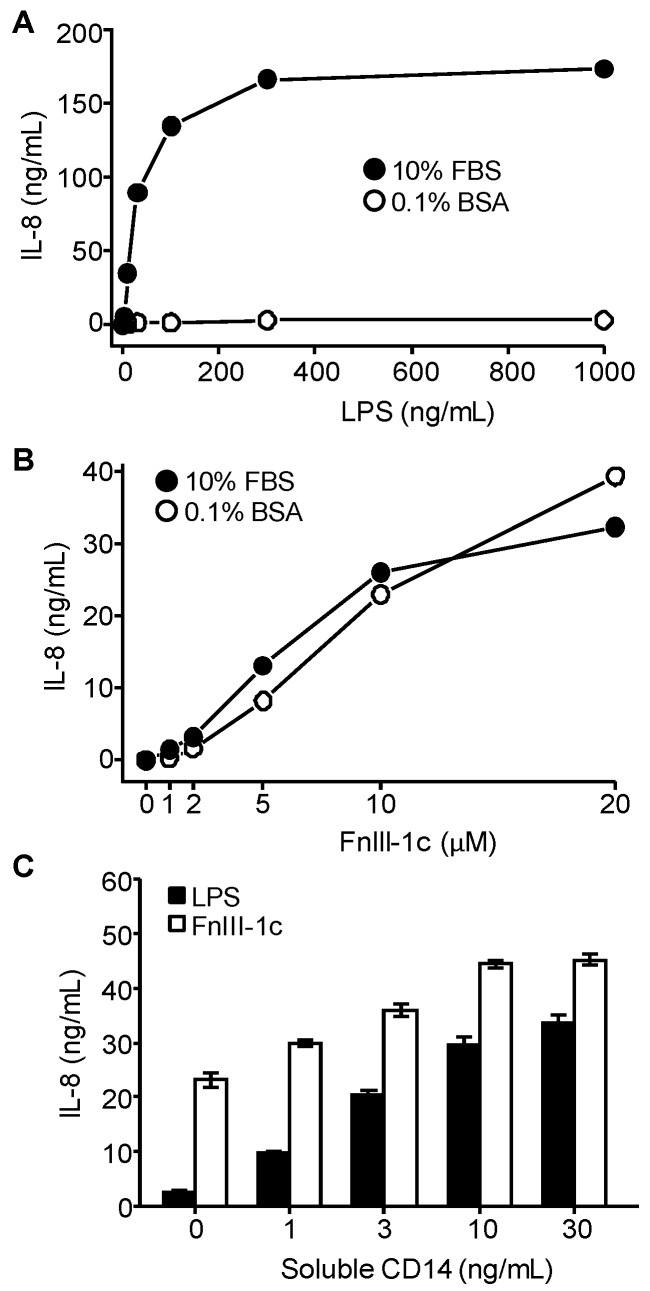
Effect of serum and exogenous CD14 on LPS and FnIII-1c-induced IL-8 expression in dermal fibroblasts. Monolayers of human dermal fibroblasts were incubated for 24 hours with the designated concentrations of (**A**) LPS or (**B**) FnIII-1c in either 10% FBS/DMEM or 0.1% BSA/DMEM. (**C**) Cells were treated for 24 hours with LPS (250 ng/mL) or FnIII-1c (10 µM) in the presence of increasing concentrations of exogenous soluble CD14 in 0.1% BSA/DMEM. IL-8 concentration in conditioned medium was measured by ELISA. The data represent the mean ± S.E. of triplicate assays from three separate experiments.

**Figure 4 cells-09-00216-f004:**
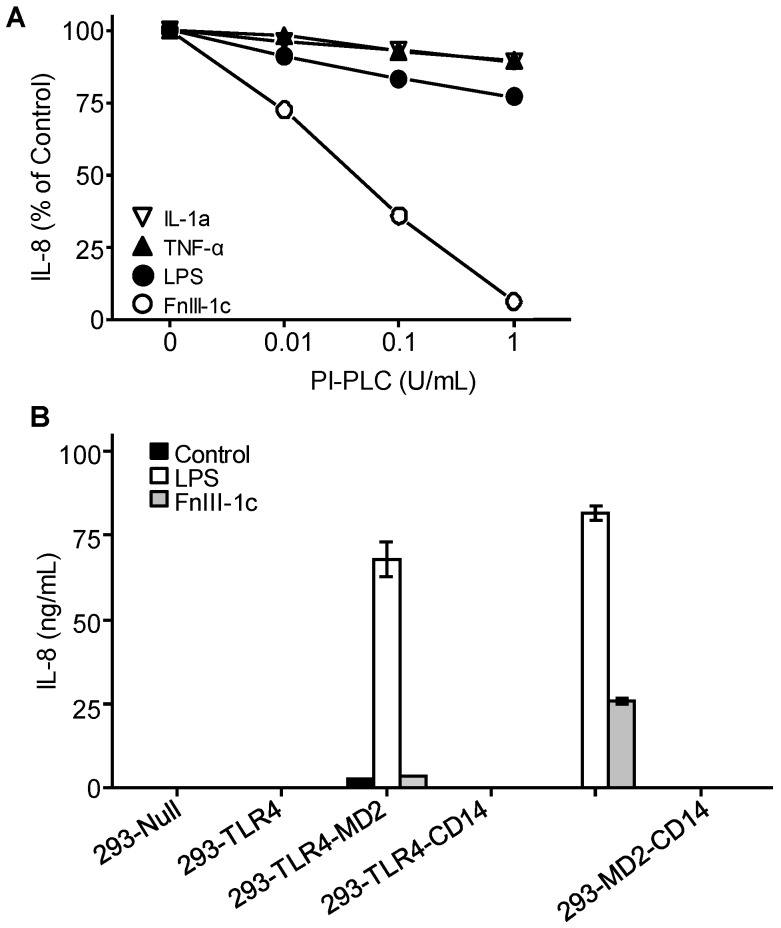
Effect of soluble and membrane CD14 on LPS and FnIII-1c-induced IL-8 expression. (**A**) Monolayers of human dermal fibroblasts were pre-incubated for 90 min with various concentrations of PI-PLC in 10% FBS/DMEM. Cells were then treated for 4 hours with IL-1α (50 ng/mL), LPS (100 ng/mL), FnIII-1c (10 µM), or TNF-α (50 ng/mL) in the continuing presence of PI-PLC in 10% FBS/DMEM. Cells which were not treated with PI-PLC were set as control (100%) (**B**) HEK-293 cells engineered to express the designated components of theTLR4 receptor complex were incubated with LPS (1 µg/mL) or FnIII-1c (20 µM) in 10% FBS-DMEM for 24 hours. Cells receiving only 10% FBS-DMEM served as control. IL-8 concentration in conditioned medium was measured by ELISA. The data represent the mean ± S.E. of triplicate assays from two separate experiments.

**Figure 5 cells-09-00216-f005:**
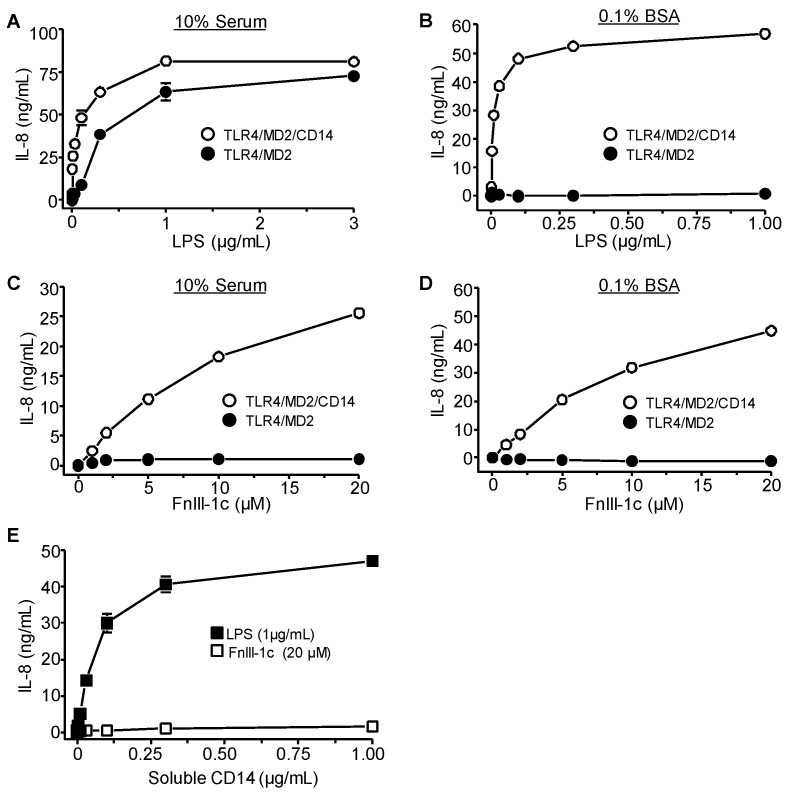
FnIII-1c-induced IL-8 expression requires membrane CD14. HEK cells expressing either TLR4/MD2 or TLR4/MD2/CD14 were incubated for 24 hours with the designated concentrations of LPS (**A**,**B**) or FnIII-1c (**C**,**D**) in either 10% FBS/DMEM (**A**,**C**) or 0.1% BSA/DMEM (**B**,**D**). (**E**) HEK-293 cells expressing TLR4-MD2 were treated with 1 µg/mL LPS or 20 µM FnIII-1c in 0.1% BSA/DMEM in the presence of the indicated concentration of exogenous soluble CD14 for 24 h. IL-8 concentration in the conditioned medium was measured by ELISA. The data represent the mean ± S.E. of triplicate assays from two (**A**–**D**) or three (**E**) separate experiments.

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
