# Peer review of "Role of TLR4 Receptor Complex in the Regulation of the Innate Immune Response by Fibronectin"

_cells, 2020, doi:10.3390/cells9010216_

Round 1
Reviewer 1 Report
Summary
FnIII-1c is peptide fragment of fibronectin. This group was the first to describe that this peptide may function as a damage associated molecular pattern molecule by activating TLR4. In this manuscript the authors investigate the activation of TLR4 by FnIII-1c. They find that similar to LPS the receptor complex of TLR4/MD1/CD14 is required for activation of TLR4 mediated pathways. Furthermore, they find that in contrast to LPS CD14 must be membrane bound for effective TLR4 pathway activation.
Major comments
The data within this manuscript are well controlled and convincing. The manuscript is clear and concise. All of the data is obtained with either human dermal fibroblasts or transfected HeK293 cells. One major limitation is that the only readout for these studies is the expression of IL-8.
It is now common for data to be presented as individual data points rather than averages in bar graphs. Also it would be appropriate to show IL-8 amount rather than % of control.
Minor comments
Figure legend of figure 3 is either truncated or IL-8 needs to be removed.
Line 216 should be respond instead of response.
Zeroes are missing from several of the graphs in multiple figures.
Possibly a +/- could be used to designate the treatments and transfections in figure 4B.
Author Response
Response to Reviewer 1:
“One major limitation is the only readout for these studies is the expression of IL8.”
Our focus in these studies was to identify which components of the TLR4 receptor complex on dermal fibroblasts were required for the activation of the innate immune response initiated by the canonical TLR4 ligand, LPS, and the fibronectin-derived DAMP, FnIII-1c. We have previously identified several TLR4 dependent fibro-inflammatory genes which are induced in dermal fibroblasts by FnIII-1c. This information has been added to the Introduction.
“It would be appropriate to show IL8 amount rather than % of control.”
We have presented most of the data as ng/ml IL8. However, the amount of IL8 released varies between ligands and from experiment to experiment. Therefore, in those experiments involving inhibitors, we have used % control to facilitate comparisons between ligands.
“Figure legend of figure 3 is either truncated or IL-8 needs to be removed”.
The legend has been corrected.
“Line 216 should be respond instead of response.”
This has been corrected.
“Zeroes are missing from several of the graphs in multiple figures”.
This has been corrected.
Reviewer 2 Report
This is an interesting study. The results generally supported the conclusion. However, a description of statistic analysis is missing.
Author Response
Response to Reviewer 2:
“A description of statistical analysis is missing.”
Data analysis is provided in each legend.
Reviewer 3 Report
The data presented in this manuscript by Zheng et al, had demonstrated the role of TLR4 receptor in fibronectin mediated regulation of innate immune response in human dermal fibroblasts. The authors have nicely presented the signaling components for FnIII-1c mediated activation of IL8 secretion in dermal fibroblasts. Overall, manuscript is well written however some experimental and conceptual problems are noted and noted below:
Most experimental data presented here are correlatives. Mechanistic data are missing. Such as how these signaling regulated IL8 secretion. Is it at transcriptional level or translational regulation. Discussion is very descriptive. it should to be more focused on mentioned signaling pathways.Minor concerns:
FNII-1c doses are inconsistence between experiments. It is always good to use the same dose for all experiments. Same with LPS and other agonist as well.
Author Response
Response to Reviewer 3:
“Most experimental data here are correlatives. Mechanistic data are missing. Such as how these signaling regulated IL-8 secretion. Is it at transcriptional level or translational regulation."
We have shown in earlier studies that in dermal fibroblasts, FnIII-1c regulates IL-8 expression at the level of transcription and message stability (Kelsh, PLoS One, 2014). We would agree that linking the components of the TLR4 complex back to specific stages of IL-8 synthesis and secretion would be an interesting future study.
“FnIII-1c doses are inconsistent between experiments. It is always good to use the same dose for all experiments. Same with LPS and other agonist as well.”
The doses are consistent for each cell type. Higher doses were used on the HEK293 cells.
Round 2
Reviewer 3 Report
None